# Feed Conversion Ratio (FCR) and Performance Group Estimation Based on Predicted Feed Intake for the Optimisation of Beef Production

**DOI:** 10.3390/s23104621

**Published:** 2023-05-10

**Authors:** Chris Davison, Craig Michie, Christos Tachtatzis, Ivan Andonovic, Jenna Bowen, Carol-Anne Duthie

**Affiliations:** 1Department of Electronic and Electrical Engineering, University of Strathclyde, Glasgow G1 1XW, UK; 2Scotland’s Rural College, Beef and Sheep Research Centre, SRUC, West Mains Road, Edinburgh EH9 3JG, UK

**Keywords:** feed intake estimation, Feed Conversion Ratio, beef production, precision livestock farming, Machine Learning

## Abstract

This paper reports on the use of estimates of individual animal feed intake (made using time spent feeding measurements) to predict the Feed Conversion Ratio (FCR), a measure of the amount of feed consumed to produce 1 kg of body mass, for an individual animal. Reported research to date has evaluated the ability of statistical methods to predict daily feed intake based on measurements of time spent feeding measured using electronic feeding systems. The study collated data of the time spent eating for 80 beef animals over a 56-day period as the basis for the prediction of feed intake. A Support Vector Regression (SVR) model was trained to predict feed intake and the performance of the approach was quantified. Here, feed intake predictions are used to estimate individual FCR and use this information to categorise animals into three groups based on the estimated Feed Conversion Ratio value. Results provide evidence of the feasibility of utilising the ‘time spent eating’ data to estimate feed intake and in turn Feed Conversion Ratio (FCR), the latter providing insights that guide farmer decisions on the optimisation of production costs.

## 1. Introduction

Livestock production is facing significant pressure globally. In addition to rising costs, farms must now focus on the challenges associated with the drive to minimise green house gas (GHG) emissions for both beef and dairy farmers since enteric methane emissions from ruminant livestock account for approximately 10–12% of global anthropogenic emissions [1]. The global human population is estimated to exceed 9 billion by 2050 and with that, meat consumption is projected to expand by more than 70% compared to 2010 [2]. To respond to the range of challenges facing the sector, the farming industry has made significant progress to improve operational efficiency. Technology has a role to play here and in recent years the dairy sector has adopted a variety of automated on-farm monitoring systems to optimise fertility and hence milk production [3,4,5,6,7,8]. These sensors enable farmers to monitor larger herds with fewer staff, thus minimising operational costs. As the market for precision livestock farming (PLF) solutions has matured, competition has forced vendors to differentiate their systems through enhancements to the range of services delivered to customers. As a consequence, many herd fertility systems now offer indications of the time spent eating and ruminating, as the basis for issuing alerts as an early warning of the onset of a potential illness and/or other key welfare events such as calving [9,10,11]. The time an animal spends ruminating is known to fall immediately before calving and significant drops in time spent eating have been validated as proxy indications of the onset of an illness condition. Such measurements provide a mechanism for early intervention in the case of illness, thus reducing recovery time and optimising productivity. Improving herd welfare has been calculated to reduce ${\mathrm{CO}}_{2}$ equivalent emission by around 9% [12].

The increasing availability of cattle-mounted sensors, collars and ear tags that produce outputs of the time that an animal spends eating has stimulated a discussion around the use of this information to estimate individual animal intake and hence feed conversation efficiency. Presently, the literature contains discussion on the genetics that may lead to efficiency, as well as how feed efficiency may be utilised in breeding programs [13,14]. It is notable that, despite the growing challenge of global food production, there has been limited research directed towards the beef sector. This would enable a step change in farming practice giving farm operatives a clear indication of which animals are performing most optimally. This could inform breeding strategies but also provide greater insight into the relationship between welfare and performance [15]. Studies have reported evidence of the relationship between time spent eating and intake with strong $R^{2}$ coefficients in excess of 0.9 [16]. The challenge however is that feed intake is highly specific to an individual animal. Davison et al. [17] have shown that Machine Learning approaches, such as the use of Support Vector Regressors or Random Forest methods, can be highly effective at using feed time measures to estimate feed intake. The present paper reports on an extension to this analysis, to determine the validity of harnessing these measures to derive an estimate of the feed intake at an individual animal level.

Within the beef sector, feed cost is the largest contributor to overall production costs (between 50% and 75%) [18]. Importing feed into a farm setting also has a carbon cost associated with it; therefore, it is highly desirable to optimise the amount of feed used. Less efficient steers consume as much as £28 more feed (per head) over a 12-week finishing period compared to the most efficient tercile of animals [19]. These costs have risen significantly in recent months as fuel and feed shortages have further driven price increases, with compound feed price for cattle rising 15% in the second quarter of 2022 [20], compared to a 2020–2021 average of 4% and 5-year average of 2%. An estimation of the efficiency of converting feed into product, i.e., beef, is highly desirable as this will inform commercial decisions that farmers make to optimise production processes, in particular the optimum time to cull and/or select animals for breeding.

The study detailed here builds on the previously reported research [8,17], which assessed the validity of using measurements of the time spent eating, in isolation and in tandem with other measurement data available in a production setting, to determine the precision of estimating feed intake. It aims to assess whether estimates of feed intake are sufficiently precise to be of value, in effect to ascertain if they can be used as a basis with which to generate an FCR [21] value with a level of inaccuracy required to segment the herd into tiers of growth performance.

## 2. Materials and Methods

### 2.1. Experimental Design, Diets and Animals

The study assessed the effectiveness of using Machine Learning approaches to estimate the FCR of beef finishing cattle fed with two different diets representative of those used commercially. Purebred Limousin sire mated with Aberdeen Angus cross-bred dam cattle, initial age 517±50 days, were used in a continuous trial with concentrate or silage-based diets. The diets (fed as total mixed rations) were generated using a diet mixing wagon and consisted of forage to concentrate ratios (g/kg Dry Matter (DM)) of either 492:508 (MIXED) or 80:920 (CONC). Steers were split equally across two diets, with each diet allocated to two pens of 20 steers each (providing 40 steers per diet). One steer was removed from the CONC group due to ill health. All steers were initially provided with the MIXED diet, with those allocated to the CONC diet adapted gradually over a 4 week period. During the adaptation period, all steers were trained to use the Hokofarm Roughage Intake Control [22] electronic feed bin (HOKO), shown in Figure 1. The HOKO feed bin has an entry gate which restricts access to a single animal at a time, opening at the presence of the animals’ eID ear tag. Each visit to the feed bin is recorded with timestamp and animal ID, along with the weight of feed at the start and end of the visit, giving record of the individual animal’s feeding behaviour. Data were exported from the Hokofarm system through weekly downloads via USB from the on-farm equipment. Throughout the trial water was provided ad libitum using a water trough.

### 2.2. Intake Prediction Models

Feed intake was recorded along with the duration of each individual visit to the feed trough. Individual daily fresh weight intakes (FWIs, kg/day) were calculated from the sum of individual visits to the trough and subsequently Dry Matter Intake (DMI) was calculated using the ratio measured from the most recent weekly diet sample. The Dry Matter Intake was used as the target of feed intake prediction models for each diet as previously detailed [17]. This provided three separate algorithm variants–proportion of feeding time relative to the group (GRP), Random Forest (RF) and Support Vector Regressor (SVR)—with each algorithm trained on the two diets.

In the present work, the objective is to assess the potential to use feed intake estimates to predict Feed Conversion Ratio (FCR). As the performances of both SVR and RF models were broadly equivalent, the analysis undertaken here focuses only on the SVR model for simplicity. Details of the model implementation are reported in prior work [17]. Daily intake prediction errors in the case of the MIXED feed ration are shown below in Figure 2. The errors are significant but are centered around 0 kg/d when aggregated over the finishing period.

### 2.3. Animal Liveweight

Animals were handled once per week, in order to capture liveweight using an animal weigh scale and to check animal sensor attachment. From these liveweight measurements, a linear regression using method of Least Squares was fit to provide Average Daily Weight Gain (kg/d).
(1)$$\mathrm{Average}\;\mathrm{Daily}\;\mathrm{Weight}\;\mathrm{Gain}=\frac{\mathrm{Average}\;\mathrm{DMI}\;(\mathrm{kg}/d)}{\mathrm{Average}\;\mathrm{Daily}\;\mathrm{Weight}\;\mathrm{Gain}\;(\mathrm{kg}/d)}$$

At the beginning of the trial, steers had a mean bodyweight of 459±60 kg (min 330 kg, max 572 kg). Over the 56 days of the trial, liveweight gain was 162.61±91.41 kg, giving final weights of 637.84±66.41 kg (min 477 kg, max 782 kg).

### 2.4. Feed Conversion Ratio Estimate Based on Intake Predictions

Feed intake data were recorded over the entire finishing period on an individual animal basis. An estimate of feed intake was derived using the SVR intake prediction model. These data sets were then used to calculate an average DMI and average predicted DMI per day of the trial. In order to make the FCR calculation throughout the course of the trial, Average Daily Weight Gain (kg/d) was calculated for each steer by fitting an ordinary Least Squares regression to weekly liveweight measurements. Ordinary Least Squares regression was used to obtain daily liveweight estimates to reduce the effect of outliers due to factors such as gut fill and water consumption around the time of weighing. Average daily intake and predicted daily average intake were then combined with Average Daily Weight Gain to calculate FCR, using Equation (Equation 2).
(2)$$\mathrm{FCR}=\frac{\mathrm{Average}\;\mathrm{DMI}\;(\mathrm{kg}/d)}{\mathrm{Average}\;\mathrm{Daily}\;\mathrm{Weight}\;\mathrm{Gain}\;(\mathrm{kg}/d)}$$

### 2.5. Herd Segmentation Based on Feed Conversion Ratio

The performance of each animal—determined via FCR—was estimated and compared to actual performance (using measured feed intake values). To illustrate the potential of this methodology within a practical context, animals were categorised using two methods: (1) into three equally sized groupings, ‘Top third’, ‘Middle third’ and ‘Bottom third’; (2) considering animals within one standard deviation of the mean FCR to be the ‘Middle’ group, with the ‘Top’ and ‘Bottom’ those that are one standard deviation above and below the mean, respectively. With segmentation method 1, each grouping contained 26 animals; with method 2, the ‘Top’, ‘Middle’ and ‘Bottom’ groups contained 8, 63 and 7 animals, respectively. The methods were applied to both the actual and predicted FCR, to allow comparison between actual and predicted performance for each animal. In this manner, we can identify the animals that gained the most and least weight per kg of feed.

Other approaches for segmenting the herd may be more appropriate, depending on the distribution of FCR that is present in a given herd, or the scale of intervention that the herdsman wishes to implement.

## 3. Results

### 3.1. Feeding Behaviours and Feed Conversion Ratio

Contrasting diets resulted in notable differences in feeding behaviours, with significantly increased fresh weight intake, number of visits and total feeding duration for the MIXED diet. When intake was converted to DMI, the mean intake and weight gain were slightly lower for the MIXED diet, at 10.4 kg vs. 11.7 kg median daily DMI and 177 kg vs. 198 kg median final body weight gain.

As described in [17], the SVR feed intake predictions exhibited a correlation with the actual fresh weight feed intake ($R_{RM}^{2}=0.24$ and 0.20, for CONC and MIXED, respectively) at an RMS error of 1.56 kg for the CONC diet and 2.44 kg for the MIXED diet (Figure 3). Closer inspection of the spread of the daily errors on any given day—illustrated by the probability density function in Figure 2—indicates that most prediction errors are <5 kg, but contain ‘outliers’ of sizeable errors.

Feed Conversion Ratio was calculated using mean intake (from measured daily intake) and average daily gain (using a Least Squares regression over body weight measured during the finishing period) as in Equation (Equation 2). Steers on a MIXED diet perform slightly worse overall than those animals in the CONC feed group, with a calculated FCR of 8.52±1.22 versus 7.53±1.40, respectively, shown in Table 1.

### 3.2. Feed Intake Modeling

Feed intake predictions in this paper are based on prior work [17]. Four models were generated on each of the MIXED and CONC diets to predict feed intake using feeding behaviours and animal metadata as input. The best performing models were the Random Forest and Support Vector Regressor, with Random Forest (RF) achieving $R^{2}$ values of 0.55 and 0.46 for the CONC and MIXED diets, and Support Vector Regressor (SVR) achieving 0.54 and 0.48 for the CONC and MIXED diets. As the RF and SVR models had similar performance in predicting feed intake, only the intake prediction results from the SVR model are used here, as this model achieved the lowest Root Mean Square Error (RMSE) across both diets, at 1.56 kg and 1.56 kg for the CONC and MIXED diets.

### 3.3. Actual vs. Predicted Feed Conversion Ratio

Using the predicted feed intake to calculate FCR results in a similar trend to that when calculating FCR from actual feed intake; animals in the MIXED dietary group have lower performance than animals in the CONC dietary group, with FCR of 8.57±1.42 and 8.03±1.86 for the MIXED and CONC diets, respectively (Table 2).

Figure 4 shows the calculated FCR plotted against the FCR, with the dashed line showing a hypothetical perfect 1:1 prediction. Strong correlations are observed for calculated and predicted FCR for both diets ($R^{2}=0.83$ (MIXED) and $R^{2}=0.93$ (CONC)).

An abbreviated table is provided in Table 3, showing the first five steers from each diet evaluated using the SVR model. The SVR models produce an error in estimating FCR of 0.02±0.54 and −0.07±0.5 for the MIXED and CONC models, respectively.

When considering animals from both diets, the Pearson correlation coefficient (*r*) between actual and predicted FCR is 0.94, with prediction error $\epsilon_{p}$ (from Equation (Equation 3)) of −0.09±0.64 (i.e., the model, on average, over-predicts by 0.09). If each feed group is considered individually, the MIXED model predicts with error −0.05±0.62, with the CONC group having an error of −0.14±0.66. Full results are presented in Table 4.
(3)$$\epsilon_{p}={\mathrm{FCR}}_{\mathrm{actual}}-{\mathrm{FCR}}_{\mathrm{predicted}}$$

### 3.4. Herd Segmentation

Predicted FCR is used to segment the animals into groups based on performance. Two approaches are demonstrated here for segmenting the animals into ‘Top’, ‘Middle’ and ‘Bottom’ groups: (1) three equal sized groups; (2) assign animals one standard deviation above and below the mean FCR as ‘Top’ and ‘Bottom’, with the remainder in ‘Middle’. These groupings are compared against their group if segmented using the actual FCR.

For method 1, in the case of the predicted ‘Top third’ category, the estimated feed intake correctly identified 19 (from 26) of the ‘Top third’ performing group. Seven of the animals were categorised as being ‘Middle third’ performers. Similarly, for animals within the ‘Middle third’ category 16 were correctly identified, 6 were classed as ‘Top third’ and 4 as ‘Bottom third’ performing. Finally, in the case of the ‘Bottom third’ category, 23 were correctly identified with 3 identified as ‘Middle third’ performers. A single animal from the ‘Bottom third’ group of animals was incorrectly categorised as ‘Top third’; no ‘Top third’ performing animals were categorised as ‘Bottom third’. Results are shown in Table 5.

As the diets are significantly different in terms of concentrate content, herd segmentation was also conducted per diet, with results in Table 6. When separated by diet, no animal is miscategorised by more than one group—two ‘Top third’ animals and four ‘Bottom third’ are classified as ‘Middle third’. No ‘Top third’ category animal was classified as ‘Bottom third’, or vice versa.

If the animals are instead segmented based on one standard deviation above or below the mean, shown in Table 7, all 8 animals are correctly identified in the ‘Top’ grouping, while 5 out of 7 are correctly identified in the ‘Bottom‘ grouping. For the ‘Middle’ grouping, 55 out of 63 are correctly identified, with 5 misclassified as ‘Top’ and 3 misclassified as ‘Bottom’. With this approach, the ‘Middle’ grouping contains many more animals, so it may be more suitable for detecting animals where performance is a notable outlier.

Again, performance groupings were also generated for each feed group, with results in Table 8. As with method 1, no animal is miscategorised by more than one group.

The availability of this loosely quantitative performance estimate, if used appropriately and within local context, can be potentially highly beneficial to farm operatives making culling or breeding choices.

## 4. Discussion

Our objective in the research presented here was to use predicted intake to estimate the performance of beef steers on an individual basis—using Feed Conversion Ratio—and group steers by performance. Intake predictions were generated from measures of cattle feeding behaviour such as number of visits to the feeder, duration per visit and steer bodyweight—variables which are available from commercial sensor systems. For example, collar systems and ear tags are able to provide a classification of time spent eating [5,23,24]. From periodic classifications of ‘eating’, number of visits and duration per visit can be derived. These can be paired with water troughs with integrated weigh scales [25] to obtain regular measures of animal liveweight—and thus liveweight gain—on an individual basis.

Cattle have historically been primarily evaluated on the basis of weight gain alone, as sensor technology to capture data at the individual level is a relatively recent development, and sensor uptake in the beef sector has lagged behind that of the dairy sector [26,27,28]. Models have previously been developed for cattle which predict DMI and residual feed intake [29,30], including models which utilise animal-mounted sensors [5,31]. Similar models have been developed for small ruminants [32]. However, these models have not been combined with animal growth data to automatically predict performance or segment the herd into categories by that performance.

With increasing commercial and environmental pressures, it is of vital importance to identify the most performant animals in a herd, to facilitate better herd management decisions. While there has been some research into predicting the body composition, feeding efficiency and growth performance of cattle [33,34,35], this typically focuses on factors such as breed and the energy value of the diet, rather than on individual animal feeding behaviours and predicted intake.

Improving cattle performance is an important aspect in reducing methane emissions and increasing sustainability [34,36,37]. With feed contributing between 50% and 75% to overall production cost in the beef sector [15], identifying efficient animals and lineages within a herd is a key part of long-term herd management decisions. This is supported by feed conversion efficiency (the reciprocal of feed conversion rate) being found to have heritability of 0.23±0.01 [13].

As a first approach to estimating steer performance within a herd, the authors have assumed that the herd can be categorised into three distinct tiers. Here, two approaches to grouping the herd have been presented; (1) splitting the herd equally into three groups, and (2) identifying animals that are more than one standard deviation away from the herd mean. In both example scenarios, using predicted intake assigned the majority of animals to their actual performance group, and when the animals were considered per diet, no individual was misclassified by more than one performance group level.

In reality, the distribution of performance within the herd may be one of numerous patterns, such as *N* distinct groups of roughly even size (as has been assumed above with N=3); *N* distinct groups, with an imbalance in size (e.g., a large group of high performing animals, with a smaller group of animals with low performance); an approximately normal distribution (most animals are ‘Middle third’, with a few ‘Top third’ and a few ‘Bottom third’ performers, as demonstrated above); an approximately linear distribution of performance across the herd; and many other potential patterns. Even with choosing *N* distinct groups, this may result in animals with very similar performances being assigned different performance groups as one falls marginally above a threshold, while the other falls marginally below. Further, the method for segmenting the herd may vary depending on the long-term strategy that the farmer desires; for example, the farmer may want to influence their herd in the direction of maximum performance and thus want to select only the best performing animals, or they may want to influence their herd in the direction of predictability and shaping the herd towards a normal distribution, and thus only removing outlier animals on the ‘Bottom third’ end of the performance spectrum. In these cases, more sophisticated approaches to identifying performance groups may be beneficial.

Clustering may be an appropriate option, with multiple variants able to target different use cases. A farmer may use partition-based clustering to identify multiple groups, such as ‘Top’ and ‘Bottom’ third of performers, and specify how granular they wish to group their herd. Alternately, fuzzy approaches could identify multiple sub-groups within the herd, guiding the farmer to look for differences in herd practice or animal information that may explain the differences.

In all cases, the farmer should inspect the statistics of each resultant grouping—such as the mean performance, and difference between groups—to identify if the groups are distinct enough to suggest any management action be taken.

## 5. Conclusions

This study quantified the inaccuracy of segmenting animals into performance groups based on estimates of feed intake based on the time spent eating. Initial results suggest that animals at the top and bottom end of the performance range can be categorised appropriately; however, there is lower confidence when categorising animals into the middling performance group. This segmentation is of value as it provides information that supports on-farm decision making, and informs future sensor developments by serving as a benchmark of what is possible when using direct measures of feed intake.

While the combined-diet model utilises more data for training, the resultant segmentation is slightly worse than when using individual feed models. This may be explained by the difficulty in learning the distinct feeding behaviour differences between the CONC and MIXED diets: on average, animals on the MIXED diet consumed around 70% more (in terms of fresh weight intake), resulting in more visits, more overall time spent eating, and a longer time per individual visit. With more variation in diet, and a larger study cohort, it may be possible to train a model which is diet-agnostic; at this stage, informing the model of the type of feed, or using a feed-specific model, would provide a better categorisation of feeding performance.

The approach adopted centred on a calculation of the scale of errors in the estimated feed intake under ideal conditions, i.e., when exact measurements of feeding time are available, most readily through electronic feeder units. However, access to such measurements is not routinely available in production settings but in the recent past, the sector has been more amenable to the adoption of neck-mounted collars and ear tags; thus, the estimate of FCR would therefore harness the time spent eating data generated by these systems. The latter data will impact the accuracy of the feed intake estimates, although that validation has yet to be executed and is the goal of future work. The research reported here is best viewed as a baseline evaluation of the feasibility of estimating feed intake as the foundation to derive Feed Conversion Ratio (FCR).

Further analysis is required to determine the monetary value to the farmer from the information that supports the more accurate identification of the optimum time to ship animals to market, e.g., the trade-off between the feed costs of marketing a slow-growing animal underweight as opposed to allocating additional time for the animal to grow. The information can also be of benefit in making breeding decisions, the subject of ongoing analysis.

## Figures and Tables

**Figure 1 sensors-23-04621-f001:**
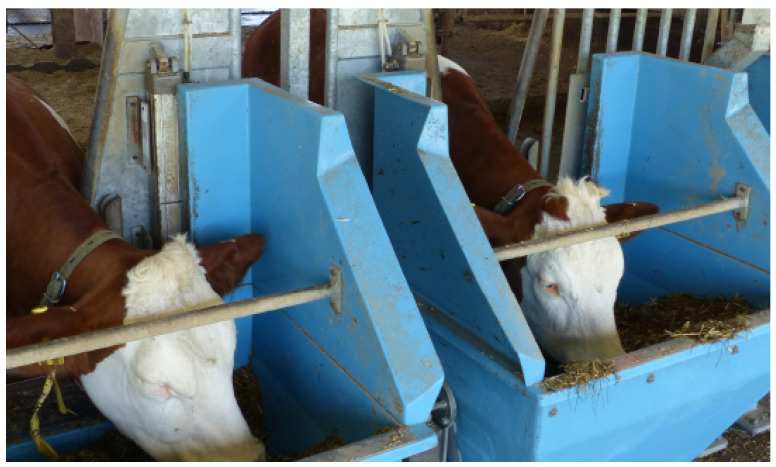
Hokofarm Roughage Intake Control electronic feed bin. Adapted under Fair Use from Hokofarm Group, 2021 [22].

**Figure 2 sensors-23-04621-f002:**
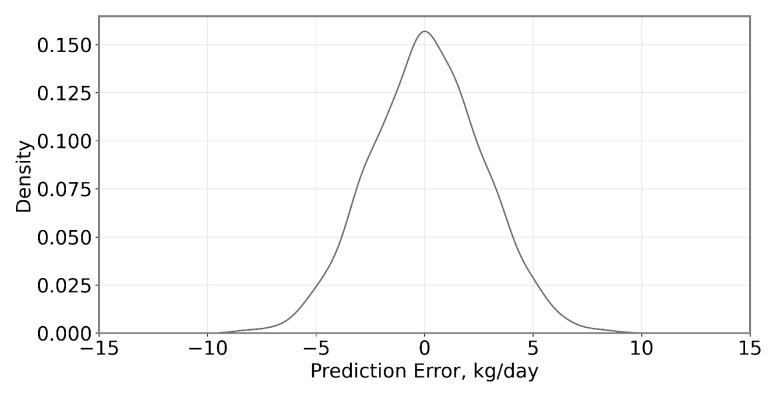
Probability Density Function (PDF) of errors in the feed intake estimate for the Forage Diet (MIXED).

**Figure 3 sensors-23-04621-f003:**
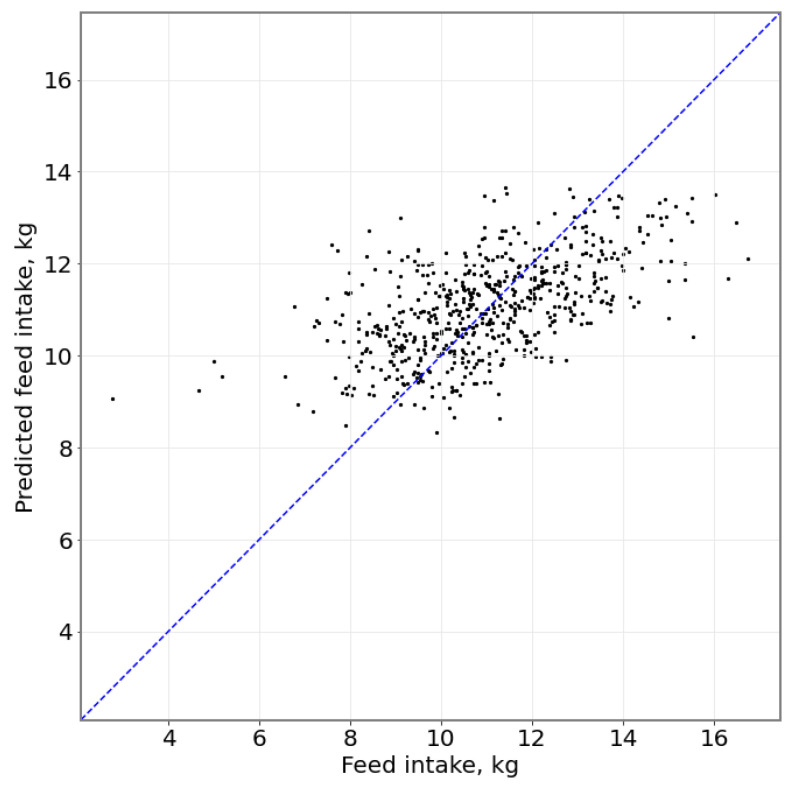
Scatter plot of actual feed intake against predicted feed intake for both CONC and MIXED feeds; $R^{2}=0.851$ with a Root Mean Square (RMS) error of 2.18 kg. Black points represents a single animal’s actual vs. predicted feed intake; blue line represents a 1:1 prediction.

**Figure 4 sensors-23-04621-f004:**
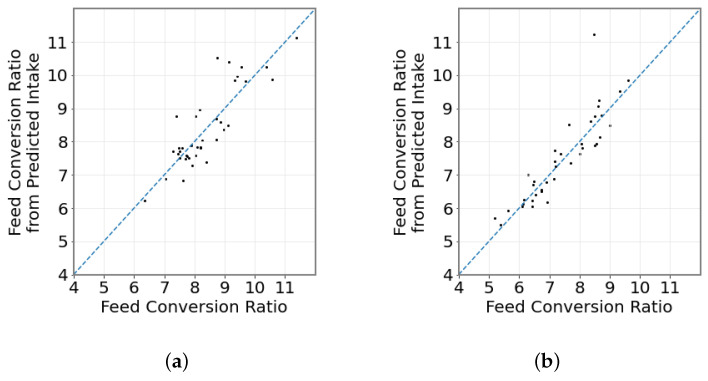
Feed Conversion Ratio (FCR) from estimated intake. (**a**) MIXED diet. (**b**) CONC diet. Black points represents a single animal’s actual vs. predicted FCR; blue line represents a 1:1 prediction.

**Table 1 sensors-23-04621-t001:** Calculated Feed Conversion Ratio.

Diet	Mean, μ	Median	Std. Dev., σ
COMBINED	8.03	8.01	1.40
MIXED	8.52	8.19	1.22
CONC	7.53	7.21	1.40

**Table 2 sensors-23-04621-t002:** Predicted Feed Conversion Ratio.

Diet	Mean, μ	Median	Std. Dev., σ
COMBINED	7.48	7.38	2.09
MIXED	8.57	7.56	1.42
CONC	8.03	7.82	1.86

**Table 3 sensors-23-04621-t003:** Prediction of Feed Conversion Ratio (sample of 5 animals from each feed).

		FCR		
Diet	Anonymised ID	Actual	Predicted	Difference
MIXED	A01	6.35	6.28	0.07
	A04	9.33	9.59	−0.26
	A05	7.49	7.86	−0.38
	A06	7.50	7.62	−0.12
	A09	8.72	8.26	0.46
		⋮		
CONC	A02	6.17	6.10	0.07
	A03	7.65	8.32	−0.67
	A07	5.18	5.59	−0.41
	A08	6.10	6.04	0.06
	A11	6.74	6.51	0.23
		⋮		

**Table 4 sensors-23-04621-t004:** Feed Conversion Ratio (FCR) prediction error, $\epsilon_{p}$ (Equation (Equation 3)).

Diet	$r^{2}$	Mean, μ	Median	Std. Dev., σ
COMBINED	0.88	−0.09	0.01	0.64
MIXED	0.81	−0.12	0.12	0.62
CONC	0.91	−0.14	−0.06	0.66

**Table 5 sensors-23-04621-t005:** Prediction of cattle feed intake performance.

Actual	Predicted Performance			
Performance	Top	Middle	Bottom	Total
Top	19	7	0	26
Middle	6	16	4	26
Bottom	1	3	23	27

**Table 6 sensors-23-04621-t006:** Prediction of cattle feed intake performance, separated by diet.

Diet	Actual	Predicted Performance			
	Performance	Top	Middle	Bottom	Total
MIXED	Top	11	2	0	13
	Middle	2	7	4	13
	Bottom	0	4	10	14
CONC	Top	11	2	0	13
	Middle	2	9	2	13
	Bottom	0	2	11	13

**Table 7 sensors-23-04621-t007:** Prediction of cattle feed intake performance, when segmenting using one standard deviation from the mean.

Actual	Predicted Performance			
Performance	Top	Middle	Bottom	Total
Top	8	0	0	8
Middle	5	55	3	63
Bottom	0	2	5	7

**Table 8 sensors-23-04621-t008:** Prediction of cattle feed intake performance, separated by diet, and segmented using one standard deviation from the mean.

Diet	Actual	Predicted Performance			
	Performance	Top	Middle	Bottom	Total
MIXED	Top	4	1	0	5
	Middle	3	28	1	32
	Bottom	0	1	1	2
CONC	Top	2	1	0	3
	Middle	3	30	0	33
	Bottom	0	1	1	2

## Data Availability

None of the data were deposited in an official repository. The data that support the findings of this study are available upon reasonable request from the corresponding author.

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
