# Peer review of "Feed Conversion Ratio (FCR) and Performance Group Estimation Based on Predicted Feed Intake for the Optimisation of Beef Production"

_sensors, 2023, doi:10.3390/s23104621_

Round 1

Reviewer 1 Report

The subject is timely and of interest to journal readers. However, the manuscript is poorly written and lacks a logical flow of ideas. The literature review is insufficient, so the reader is not provided with sufficient background information on the topic. There are several points that require revision.

The wide variety of problems that plague the farming industry, the industry has made great strides in minimizing gas emissions and estimating feed intake using predicted feed intake.

We think this article needs a better title. If the study's indicators were included in the title, readers would have a better idea of what to expect.

The manuscript's scholarly accuracy should add to the subject area compared with other published material, and the predict the feed conversion ratio is an area that has not been extensively explored in previous works.

The authors should consider regarding the methodology the specific improvements by following :

Three distinct algorithm variants, proportion of feeding time relative to the group, Random Forest (RF), and Support Vector Regression, each trained on the two diets, are regarded as effective by other researchers.

The conclusions was consistent with the evidence and arguments presented but too long, which made expressing the objective achievement difficult.

The reference relevant and appropriate to this work.

Reviewer 2 Report

The present work presents a study on the estimation of the feed conversion ratio (FCR) based on the predicted feed intake for the optimization of beef production. The subject is interesting and particularly important, namely in terms of the need to control the evolution of animal consumption in line with the evolution of their body mass. Although the present work is subsequent to two works by the authors, references 14 and 17, there are several flaws at the structural level of the work that require revision, namely:

- There is no state of art that is necessary to support the study, the reference to previous works is not enough;

- An exhaustive description of the technologies used is required (Sensors used, LoRa Equipment, RFID, etc.) and how the data is read and sent to the CLOUD, Server, LoRaWAN Gateway, or other technology, integrated with the TCP network /IP, and what type of communication architecture is used.

Reviewer 3 Report

The Introduction section is not sufficient.

The method section is not sufficient

1. Line 41: The first sentence should be re-written. Such as “Livestock production is face to face significant problems” or like this.

 2. Line 61: “CO2” should be “CO2”.

3. Line 77: “feed accounts” should be “feed expenses”.

4. Line 83: “in Q2 2022” it is so hard to understand at first reading.

5. Line 118: “Random Forest (RF) and Support Vector Regressor”. The authors mentioned that both RF and SVR were used. But there is nothing in abstract about RF.

6. Results don’t include any information about so important goodness of fit criteria such as RMSE, RAE, MAPE so on for SVR. Nothing is given about sensitivity analysis for support vector regression which is so significant for this algorithm.

Reviewer 4 Report

Feed conversion ratio is an essential indicator to assess the performance and efficiency of beef production. This manuscript estimates the individual animal feed intake to predict the feed conversion ratio (FCR), and use this information to categories animals into three groups (‘good’, ‘average’ and ‘low’). This is meaningful in guiding farmer decisions on the optimization of production costs. The authors need to justify how did they define of each category. Additionally, please explain more about how to predict the feed intake (eg. What’s the input or key hyper-parameters used to predict DMI), even though this may be detailly reported in prior work.

Line 99: what’s the initial bodyweight and age of these experimental steer?

Line 125: these are two “are”, delete one of them.

Section 2.3: how often and how long to measure the bodyweight of cattle?  what method used to get the bodyweight?

Section 2.4: suggest to move this section to the results part.

Section 2.5: suggest to find an objective way to define the category as ”good”, “average” and “low”. For example, whether there is a threshold for FCR to see the performance is good or not, based on the knowledge of commercial beef production. There is no need to satisfy the rule of equal size of each defined category.

Line 164-165: “570.0 kg vs 575.4 kg median final body weight gain.” please check, this is the body weight gain or the final body weight over 56-day period?

Figure 3: suggest to use the same range in horizontal and vertical axes.

Line 181-182: the R2 is very low, in the range of 0.48-0.55. Did you try to optimizing the model? eg. Hyper-parameter optimization, k-fold cross validation.

Round 2

Reviewer 1 Report

The revised manuscript has been constructed and written in an appropriate manner. The formatting for the manuscript is correct throughout each section. Due to the precision of the work and the skill with which it is presented, the manuscript satisfies the criteria necessary for publication in your Journal.

Author Response

The authors would like to thank the reviewer for their constructive comments and feedback regarding the submitted manuscript.

Reviewer 2 Report

The present work presents a study on the estimation of the feed conversion ratio (FC) based on the prediction of feed intake to optimize beef production. As mentioned in the previous review, the subject is interesting and particularly important, namely in terms of the need to control the evolution of animal consumption according to the evolution of their body mass.

Constructive recommendations were made regarding the need for an exhaustive description of the technologies used, however, the authors limited themselves to adding two paragraphs without relevance to the subject, since no bibliographic reference was even included.

The work is still interesting, however, in this way, without an exhaustive description of the technologies used and properly referenced, the article does not fit a publication for the scientific journal sensors, even in a special edition "Sensors and Data-Driven Precision Agriculture", perhaps for another scientific journal in the field of animal production. Collecting data with a USB stick takes us back many years.

Author Response

We thank the reviewer for the additional comments. We agree that the subject area has significant interest and importance to assess  animal consumption and efficiency. 

This paper uses a direct measurement of feeding duration and number of visits to predict feed conversion ratio. Other technologies eg collar or ear tag can estimate feeding duration but there is an error associated with this. In the present experiment we have removed this error by using a direct measurement which allows us to gauge what the best possible  performance would be. The hardware technological solution (Hokofarm Roughage Intake Control) is not novel, but processing the sensor data is.

On that basis, and given the interest and importance of the subject area, we assert that the paper makes contributions to the processing of sensor data and is underpinned by the data-driven nature of the proposed methodology which would be of interest to the journal readership. 

We thank the reviewer for their thoroughness in identifying the missing citation. The Hokofarm feed weighing system mentioned in the methodology has been updated to include reference to the product website, both in the bibliography and cited in the text. The eID ear tags are commodity tags used within standard farming practice in the UK, and do not impact the presented analysis.

Reviewer 3 Report

The Discussion section should be expanded by comparing related literature results.

Author Response

There are a number of papers in the literature that relate to the correlation of feeding behaviours with feed intake, including commercial systems and research which measure feeding behaviour. However there has been little consideration in the literature to the subsequent step of predicting animal performance, which is a key factor in farm management practice decisions.

We have expanded the discussions sections to include information on the current state of intake and performance prediction, and the importance of selective breeding. We thus identify the contribution we believe our paper makes to the literature.

Reviewer 4 Report

The authors responded and justified all points of the review. Please make sure to proofread the manuscript again.

Author Response

The authors would like to thank the reviewer for their constructive comments and feedback regarding the submitted manuscript. The manuscript will be proofread prior to final submission.